# Global Investigation of Cytochrome P450 Genes in the Chicken Genome

**DOI:** 10.3390/genes10080617

**Published:** 2019-08-14

**Authors:** Junxiao Ren, Liyu Yang, Quanlin Li, Qinghe Zhang, Congjiao Sun, Xiaojun Liu, Ning Yang

**Affiliations:** 1Department of Animal Genetics and Breeding, College of Animal Science and Technology, China Agricultural University, Beijing 100193, China; 2National Engineering Laboratory for Animal Breeding, and Key Laboratory of Animal Genetics, Breeding and Reproduction, Ministry of Agriculture and Rural Affairs, China Agricultural University, Beijing 100193, China; 3College of Animal Science and Veterinary Medicine, Henan Agricultural University, Zhengzhou 450002, China

**Keywords:** Cytochrome P450, phylogenetic tree, gene structure, CYP2AC1, chicken

## Abstract

Cytochrome P450 (CYP) superfamily enzymes are broadly involved in a variety of physiological and toxicological processes. However, genome-wide analysis of this superfamily has never been investigated in the chicken genome. In this study, genome-wide analyses identified 45 chicken CYPs (cCYPs) from the chicken genome, and their classification and evolutionary relationships were investigated by phylogenetic, conserved protein motif, and gene structure analyses. The comprehensive evolutionary data revealed several remarkable characteristics of cCYPs, including the highly divergent and rapid evolution of the cCYPs, and the loss of cCYP2AF in the chicken genome. Furthermore, the cCYP expression profile was investigated by RNA-sequencing. The differential expression of cCYPs in developing embryos revealed the involvement of cCYPs in embryonic development. The significantly regulated cCYPs suggested its potential role in hepatic metabolism. Additionally, 11 cCYPs, including cCYP2AC1, cCYP2C23a, and cCYP2C23b, were identified as estrogen-responsive genes, which indicates that these cCYPs are involved in the estrogen-signaling pathway. Meanwhile, an expression profile analysis highlights the divergent role of different cCYPs. These data expand our view of the phylogeny and evolution of cCYPs, provide evolutionary insight, and can help elucidate the roles of cCYPs in physiological and toxicological processes in chicken.

## 1. Introduction

Cytochrome P450 (CYP) superfamily enzymes are broadly involved in a variety of physiological and toxicological processes, such as metabolism of endogenous and exogenous molecules, and host defense responses [1,2,3,4]. Therefore, CYPs are of particular relevance for clinical pharmacology and are of interest to many researchers. Since rat *CYP2B1* and *CYP2B2* sequences were first fully published in 1982, CYPs have been identified in various biological phyla. Recently, it was reported that there are more than 13,000 named CYPs in animals and over 16,000 in plants [3]. Because CYPs are a gene superfamily, their nomenclature system is based on hierarchical clustering of genes into subfamilies [5]. Seventeen subfamilies were found in the human genome, and 11 subfamilies were found in zebrafish [5]. However, typical angiosperm plant genomes have about 300 CYPs classified in about 50 subfamilies [1,2,6]. Nelson predicted that we may soon know of over 100,000 plant CYPs and 50,000 animal CYPs [3]. Therefore, it is difficult to develop a comprehensive understanding of the existence and overall relationships of CYPs. 

Neither the evolutionary relationships of the CYPs, nor the expression characteristics of the superfamily have been well studied. Recently, a global profiling of physiological CYP mRNA expression in multiple canine tissues was performed using RNA-sequencing (RNA-seq); the results showed that different CYPs exhibited significant tissue-specific distributions [7]. Moreover, CYP expression is under the transcriptional control of multiple mechanisms and factors, such as genetic profiles, xenobiotics, cytokines, hormones, metabolic challenges, diet and disease states, and age [8,9,10,11,12]. Many of the factors are inducers or inhibitors for different CYP expression in different organs [13]. Metabolism of sex hormones, such as 17β-estradiol, was also shown to be related to CYPs. For example, CYP1A1, which is considered one of the most important CYPs, was shown to be responsible for 2-hydroxylation of 17β-estradiol in extrahepatic tissues [14,15]. Recently, CYP1A1 was reported to be significantly increased by 17β-estradiol in MCF-7 clonal variant cells [16]. Further research is needed to determine the global expression of CYPs under estrogen administration. 

Birds exhibit substantial variation, such as in feeding habits, specific adaptations, and worldwide distributions. Therefore, birds are exposed to a variety of xenobiotic compounds, such as drugs and environmental chemicals [8,17]. In addition, birds are valuable for studying genome and gene evolution because they are evolutionarily positioned between mammals and lower orders. Studies of CYPs in avian species will provide novel insight into avian xenobiotic metabolism and the CYP family. However, knowledge of identity and expression characteristics of avian CYP genes is quite limited for the CYP1–3 subfamilies. Members of the CYP1–3 subfamilies are the major xenobiotic-metabolizing enzymes, and are primarily expressed in the liver [18,19]. Recently, a few publications systemically studied CYP1–3 subfamilies of birds [4,5,6,7], and expanded our understanding of the relationships and functional roles of the avian CYP1–3 subfamilies. However, further understanding of the whole avian CYP system is needed.

Given the importance of CYPs in the metabolism of drugs and xenobiotics, we carried out a systematic analysis of global chicken CYPs (cCYPs) for the first time. In this study, we elucidated the existing isoforms of the cCYPs and assessed the evolutionary relationships of the cCYP family. We identified 45 cCYPs in the chicken genome, and further investigated their phylogeny, gene structure, and conserved motifs. Subsequently, the expression of all cCYPs in developing embryos and livers were investigated by RNA-seq. In addition, we also investigated the effects of 17β-estradiol on all cCYPs in chicken liver by RNA-seq. Our results provide important information on CYP phylogenetic relationships and a global overview of cCYP expression.

## 2. Materials and Methods 

### 2.1. Ethics Approval

All animal experiments were approved by the Animal Welfare Committee of China Agricultural University (permit number XK622) and performed in accordance with the protocol outlined in the “Guide for Care and Use of Laboratory Animals” (China Agricultural University).

### 2.2. Animals

The chickens used in this study were pure lines of LuShi chicken, and the embryos used were pure lines of White Leghorn. The eggs were disinfected and incubated in an automated egg incubator at 37.5 °C and 65% relative humidity, with rotation every 6 h. All chickens were raised in the same environmental conditions with food and water ad libitum. 

### 2.3. Identification of P450 Members

The cytochrome P450 protein domain (PF00067, http://pfam.xfam.org/family/PF00067#Animals) was used to identify the CYPs in human, chicken, and zebrafish genomes. All CYPs were obtained from Ensembl using BioMart searching (http://www.ensembl.org/biomart/martview/) for genes that matched the PFAM PF00067 domain. The reference genome used was human genome assembly GRCh38, chicken genome assembly galGal5 and zebrafish genome assembly GRCz11. Then, all of the genes were used as queries to perform multiple database searches against the proteome and genome. BLASTN and BLASTP tools available in NCBI (https://blast.ncbi.nlm.nih.gov/Blast.cgi) were then used to identify all CYPs. The symbol names of CYPs were annotated by integrated analysis of the EBI and NCBI databases. Additionally, many genes harbored more than one transcript and/or protein; therefore, for convenience, the longest protein sequence and the corresponding transcript sequence of each gene were extracted and employed in this study.

### 2.4. Gene Characteristics and Sequence Analyses

All filtered protein sequences were downloaded with BioMart and aligned with Clustal X2.0. Gaps and missing data were excluded from the analyses by partial deletion with a site coverage cutoff of 95%. Then, phylogenetic trees based on the protein sequences were constructed using the neighbor-joining method in MEGA5. The reliability of the trees obtained was tested using bootstrapping with 1000 replicates. Conserved motifs were discovered using the MEME program (http://meme.sdsc.edu/meme/intro.html), with an optimum motif width of 6–50, a maximum of 10 returned motifs, and a minimum of 10 sites for each motif. The gene structures were analyzed with GSDS (http://gsds.cbi.pku.edu.cn/) by comparing coding and genomic DNA sequences of each gene [20].

### 2.5. RNA-seq Library Construction and Sequencing

To investigate the dynamic expression profiles of cCYPs in developing embryos, embryos were collected at 24-h intervals and labeled E1–E8. Three biological replicates for each group were collected and used to perform RNA-seq. Total RNA was extracted using Trizol (Invitrogen, Carlsbad, CA, USA). The RNA concentration and integrity were measured using a NanoDrop 2000 Spectrophotometer (Thermo Fisher Scientific, Wilmington, DE, USA) and Agilent 2100 Bioanalyzer (Agilent Technologies, Santa Clara, CA, USA). The Ribo-Zero RNA-seq libraries were prepared using the Ribo-Zero rRNA Removal Kit (Epicentre, Madison, WI, USA) and NEBNextR Ultra™ Directional RNA Library Prep Kit for IlluminaR (New England Biolabs, Ipswich, MA, USA). The prepared libraries were sequenced with the Illumina HiSeq 4000 platform using the 150-bp pair-end sequencing strategy. 

To investigate the estrogen-responsive cCYPs, four groups (*N* = 6 for each group) of 10-week-old chickens were intramuscularly injected with 0, 0.5, 1.0, or 2.0 mg of 17β-estradiol (Sigma, St. Louis, MO, USA) (dissolved in olive oil)/kg of body weight. The chickens were euthanized after 12 h, and the livers were collected. Of the harvested livers, three livers with 0 mg (Con) and three livers with 2.0 mg (E2.0) of 17β-estradiol were used to perform RNA-seq. The library preparation protocols were the same as described above. 

To investigate the expression profile of cCYPs in different developmental stages of chicken livers, pre-laying (20 weeks old) and laying hen (30 weeks old) livers were collected and RNA-seq was performed. The detailed protocols were described in our previous publications [21].

All RNA-seq data generated have been deposited in the NCBI Sequence Read Archive under the accession number SRA660677 and are included in the published article [21].

### 2.6. Sequencing Data Processing

Raw reads were cleaned using the FASTX-Toolkit (http://hannonlab.cshl.edu/fastx_toolkit/). Clean paired-end reads were aligned to the chicken genome (Galgal 5) using TopHat2 [22]. HTSeq [23] was used to count the numbers of reads mapped to each gene. Gene expression levels were normalized using fragments per kilobase of exon per million mapped reads (FPKM) values. Cuffdiff [24] was used to analyze the differentially expressed genes. Genes with |log2^fold changes^| ≥ 1 and FDR ≤ 0.05 were identified as differentially expressed genes. 

### 2.7. Quantitative Real-Time PCR (qRT-PCR)

One microgram of the total RNA from each sample was reverse-transcribed into cDNA using the PrimeScript™ RT Reagent Kit with gDNA Eraser (TaKaRa, Dalian, China) according to the manufacturer’s protocol. The cDNA was stored at −20 °C until use. The mRNA expression levels were quantified by a qRT-PCR assay, which was carried out using the SYBR Green method and performed on a Roche Lightcycle R96 instrument (Roch, Applied Science, Indianapolis, IN, USA). All reactions were carried out in a 20-μL volume that contained 2 μL (500 ng/μL) of cDNA, 10 μL of SYBR Green PCR Master Mix (Takara), 0.7 μL each of forward and reverse primers (10 μM), and 6.6 μL of RNase-free water. The reaction conditions were as follows: One cycle of pre-denaturation at 95 °C for 5 min and 40 amplification cycles (95 °C for 20 s, 60 °C for 30 s, and 72 °C for 20 s). All reactions were carried out in triplicate, and data were normalized to cycle threshold values for β-actin using the 2^−ΔΔCt^ method. Significant differences between groups were analyzed via one-way ANOVA. The primers were designed using Primer-BLAST (https://www.ncbi.nlm.nih.gov/tools/primer-blast/). The primers sequences are listed in Appendix A.

## 3. Results

### 3.1. CYP Identification and Phylogenetic Analysis

The BioMart tool was used to obtain the sequences that contained cytochrome P450 domain (PF00067). In total, 57, 45, and 82 CYPs were identified in human, chicken, and zebrafish genomes, respectively. The different number of CYPs among the three genomes indicated that dynamic gain or loss of CYPs occurred during vertebrate evolutionary history. To investigate the evolutionary relationships of the CYPs, a phylogenetic tree based on neighbor-joining method was constructed based on the protein sequences (Figure 1). All 184 CYPs were grouped into five clusters (I, II, III, IV, and V) based on the phylogenetic tree results. As expected, each CYP subfamily was completely grouped into a single cluster. For example, all 73 CYP2 family genes clustered into one branch, which was labeled cluster II. The three human, chicken, and zebrafish CYP19A1 members clustered into the smallest branch, which was labeled cluster III. Cluster I included CYP1, CYP17, and CYP21 subfamilies, whereas cluster V included CYP11, CYP24, and CYP27 subfamilies. Cluster IV included the other nine subfamilies (CYP3, 4, 7, 8, 20, 26, 39, 46, and 51). Interestingly, most CYP2 subfamily members did not have homologous genes. For example, CYP2X and CYP2AA genes are only present in the zebrafish genome, and these genes clustered in one branch. A similar situation was also observed in zebrafish CYP2k genes. Moreover, most members of the human CYP2 subfamily (11/15 members) clustered into one branch, which indicated that no homologous genes of human CYP2 genes were identified in chicken or zebrafish genomes. We found that there were as many as 39 CYP2 subfamily members in the zebrafish genome, but only 15 members in the human and chicken genomes. This finding indicates that many CYP2 subfamily members were lost during evolution. 

### 3.2. Phylogenetic and Gene Characteristic Analyses of cCYPs

To better understand the evolutionary relationship among cCYPs, a phylogenetic tree of the 45 cCYPs was constructed (Figure 2). The phylogenetic tree was also divided into five clusters, and the results were very similar to the cluster classification results shown in Figure 1. Subsequently, the exon–intron structures of the genes were investigated to elucidate the gene structural evolution (Figure 2). The results showed that cCYP1C1 was the shortest (1.2 kb) and had only one exon. In contrast, cCYP46A1 had the most exons (15), and cCYP7A1 was the longest gene (12.8 kb). For the cCYP1 subfamily, there were four members with three different exon numbers, which was also observed with the cCYP26 subfamily. However, most cCYP2 subfamily members (13/17) had nine exons. The constant exon numbers in the cCYP2 subfamily supports their close evolutionary relationship and further indicated that the CYP2 subfamily may have undergone gene duplication events during evolutionary history. Chromosomal location analyses showed that 42 cCYPs were distributed in 13 chromosomes, and the other three cCYPs were located on three scaffold regions (Appendix A). Interestingly, we found that most cCYPs (39/45) were located on chromosomes 2–10 and 14. Only cCYP2D6 was located on chromosome 1, which was the longest chromosome in the chicken genome. Moreover, no cCYPs were found to be located on macrochromosome Z. These data strongly suggested that cCYP distributions were not random. Overall, the gene characteristic investigation revealed high divergence of cCYPs.

### 3.3. Conserved Motif Analysis of cCYPs

To further elucidate cCYP evolution, the MEME program was used to identify the conserved motifs in protein sequences. The distribution of the 10 most enriched motifs is shown in Figure 3, among which, motifs 1–4 were annotated as the cytochrome p450 domain. All of the identified cCYPs contained motif 3, indicating that all of the identified cCYPs were typical of the P450 family. Results showed that genes in one cluster generally have the same motifs. For example, motifs 1–5 were conserved in clusters I, II, III, and IV, with the exception of cCYP1C1. Motif 9 was conserved in clusters III and IV, but was not identified in other clusters. Additionally, the majority of cCYPs contained motifs 1–5 with some exceptions. Specifically, motifs 8 and 9 were particularly conserved in clusters III and IV. In addition, along with all of the conserved motifs, most cCYP2 subfamily members have the specific motif 10 at the N-terminal region. Taken together, the conserved motifs and the sequential order of these motifs in the same cluster showed a high similarity.

### 3.4. Expression Profile of cCYPs in Developing Chicken Embryo

The sensitivity of developing embryos to xenobiotics is highly dependent on CYP expression [25]. Thus, cCYPs expressions in whole embryo preparations were assayed at 1–8 days of incubation by transcriptome analysis (Figure 4a and Appendix A). Results showed that all 45 cCYP genes were expressed (FPKM > 0.1) at the detected embryos; but these genes (cCYP2AB1, cCYP2AB5, cCYP2A13, cCYP1C1) were expressed at extremely low levels. We found that cCYP51A1 had the highest mean expression level in the embryos, followed by cCYP20A1, cCYP2AC1, cCYP2J23, and cCYP27C1L. As Figure 4 shows, most of the cCYPs were not steadily expressed across the different stages of chicken embryo. Significantly, cCYP1B1, cCYP2AC1, and cCYP2D6 showed gradual increases across embryo development. In contrast, cCYP20A1 and cCYP26A1 were observed to gradually decrease. 

### 3.5. Expression Profile of P450 Members in Chicken Liver

The chicken liver, one of the most important metabolic organs, synthesizes massive amounts of protein and lipids to meet the needs for egg-yolk formation during the laying period. Therefore, there exists a huge difference in the metabolic level of the livers between pre-laying and laying hens. To understand the potential of cCYPs in chicken liver metabolism, we constructed cCYP expression profiles in different stages (20 week old pre-laying hens and 30 week old laying hens) of chicken liver by transcriptome analysis. As results, we identified 31 expressed cCYPs (FPKM > 0.5) and their expression profiles were shown in Figure 4b. Eleven of them were differentially expressed (Table 1; |log2fold change| > 1 and false discovery rate (FDR) < 0.05). These 11 differentially expressed genes were validated by qRT-PCR with the exception of cCYP2B4L (Figure 5a). Notably, all differentially expressed genes were upregulated in the liver of laying chickens compared with the liver of pre-laying chickens with the exception of cCYP2AC1; cCYP2AC1 was downregulated about five-fold. During the laying stage, gene expression is highly stimulated in the liver to support the metabolic changes associated with the development of the reproductive organs [21]. Significantly regulated cCYPs in the liver indicated these cCYP members may play an important role in liver metabolic processes.

### 3.6. Identification of Estrogen-Responsive cCYPs 

It was previously reported that some CYPs appeared to be responsible for the metabolism and toxicity of estrogens [14,15,26]. Here we identified the estrogen-responsive cCYPs members in chicken liver by transcriptome analysis. As results, we identified 31 expressed cCYPs (FPKM > 0.5) and their expression profiles were shown in Figure 4c. By comparing the analysis, 11 differentially expressed cCYPs (|log2fold change)| > 1 and FDR < 0.05) were identified as estrogen-responsive genes (Table 2). The 11 differentially expressed genes were validated by qRT-PCR with the exception of cCYP2W1 (Figure 5b). Among the 11 estrogen-responsive cCYPs, 50% (six genes) were cCYP2 subfamily members. Moreover, all estrogen-responsive cCYPs except cCYP51A1 and cCYP7A1 were suppressed by 17β-estradiol administration. 

Additionally, we found that the expression of homologous genes always have different patterns. For example, no homologous pairs were identified as estrogen responsive genes. Moreover, cCYP2AC1 always has relatively high expression levels in developing embryos and livers. However, the expression levels of cCYP2AC2 were relatively low in these samples. Similar expression patterns were observed in cCYP2J23/cCYP2J2, cCYP1A2/cCYP1A1, cCYP2W1/cCYP2W2 and cCYP3A4/cCYP3A5 homologous pairs. These results further suggest that homologous genes have functional divergence.

### 3.7. Expression Characteristics of cCYP2AC1

Next, we focused on cCYP2AC1, which was one of the most changed genes in our transcriptomic data. As shown in Figure 1, the cCYP2AC subfamily was only identified in the chicken genome, but not in human and zebrafish, and had the closest evolutionary relationship with the CYP2K subfamily. It’s important to point out that cCYP2AC1 was also known as cCYP2K1L in the NCBI database. The CYP2K1L gene is only conserved in dogs, zebrafish, and frogs. Moreover, a pseudogene CYP2AC1 was found in the human genome, which suggests that the CYP2AC subfamily may have lost their functions in humans during the evolutionary process. To better understand the function of cCYP2AC1, we first characterized the tissue distribution of cCYP2AC1. Results showed that cCYP2AC1 was primarily expressed in the kidney and liver in both 20- and 30 week old chickens (Figure 6a,b). Then, the mRNA expression levels of cCYP2AC1 in the liver during the different developmental stages of chicken were studied. The highest expression level was detected in the liver of 5 week old chickens (Figure 6c). In contrast, the lowest expression level was detected in the liver of 30 week old chickens (Figure 6c). To confirm the effect of 17β-estradiol on the mRNA expression of cCYP2AC1, different concentrations of 17β-estradiol were administered in vivo. As a positive control, the mRNA expression level of APOV1, which is strictly dependent on estrogen [27,28,29], significantly increased after 17β-estradiol administration (Figure 6d). As a result, intense suppression of cCYP2AC1 expression levels was observed in the 17β-estradiol-treated groups compared with the control groups (Figure 6e).

## 4. Discussion

CYP enzymes are broadly involved in a variety of physiological and toxicological processes. Investigation of cCYP characteristics and expression profiles will further elucidate cCYP functions. To date, various CYPs have been identified and functionally characterized in various organisms [3]. However, less information is available for bird CYPs, especially regarding global identification and expression profiles of cCYPs. In this study, both BioMart and BLAST tools were employed to identify global cCYPs in the chicken genome. Forty-five cCYPs were identified from the chicken genome, which is less than the number of CYPs in human and zebrafish genomes. The fewer cCYPs in the chicken genome indicate that some cCYPs were lost during bird evolution. 

A controversial CYP gene is CYP2AF. CYP2AF was previously hypothesized to have been lost in avian lineages [8]. However, CYP2AF was then reported to be found in 26 avian species but not in the chicken genome [17]. In our study, no cCYP2AF members were identified, which indicates loss of CYP2AF in the chicken genome. Moreover, the cCYP2 subfamily members identified in our study are consistent with those reported by a previous study, except for the cCYP2H gene [17]. The chicken CYP2H subfamily, also known as CYP2C23, has been widely studied [30,31,32,33]. However, our phylogenetic analysis showed that “cCYP2C23/cCYP2H” is embedded in the CYP2C branch (Figure 1). Synteny analysis also showed that CYP2H is an ortholog of CYP2C62P in humans, CYP2C23 in rats, and CYP2C44 in mice [34]. *Kubota* et al. suggested that CYP2H subfamilies in birds should be renamed CYP2C23 [34]. Standardized gene nomenclature is necessary to understand the evolutionary relationships among CYPs. Therefore, cCYP2C23 was shown to be used instead of CYP2H in chicken. These results indicate that birds have likely developed an array of novel CYP systems and xenobiotic-metabolizing mechanisms. 

To better evaluate the evolutionary relationships of CYPs, we investigated the phylogenetic relationships among the 184 CYPs identified in human, chicken, and zebrafish genomes. We grouped all 184 members into five clusters based on their phylogenetic relationships (Figure 1). This classification is helpful for understanding the relationship of the different CYP subfamilies. For example, the CYP24 and CYP27 subfamilies were grouped in cluster V; it was previously reported that CYP27B1 and CYP24 were vitamin D-metabolizing enzymes and have been well studied [35,36,37,38]. Likewise, cluster I included three different CYP subfamilies: CYP1, CYP17, and CYP21. Interestingly, CYP17 and CYP21 were shown to play critical roles in steroid hormone biosynthetic pathways and are coordinately regulated by ACTH [39,40,41]. Moreover, CYP17 catalyzes steroid 17a-hydroxylase and 17–20 lyase activities at key points in estradiol biosynthesis [42,43], and CYP1A1 is also considered a major enzyme that participates in estrogen hydroxylation [44,45,46,47]. Therefore, cluster I members were suggested to have functions in the steroid hormone signaling pathway. 

We constructed a phylogenetic tree of the 45 cCYPs and grouped these genes into five clusters. Despite some modest differences, the classification was relatively consistent with the classification in Figure 1. As a kind of evolutionary relic, gene structure carries the imprint of the evolution of a gene family [20,48,49]. Structural analyses can provide valuable information on gene evolutionary relationships. Gene structure analysis showed that cCYPs occupied various exon–intron structures (Figure 2). For example, the genes in cluster III were the shortest, although a greater degree of variation in exon number existed in this cluster, which varied from 1 to 11. Interestingly, it is remarkable that the gene structure of cCYP2 subfamily genes (cluster IV) were uniformly distributed in exon–intron structure, which further indicated that gene duplication events occurred in the cCYP2 subfamily. For conserved motif analysis, we found that conserved motifs often existed in the same cluster, which further supported the phylogenetic classification of cCYPs. Overall, gene structure and conserved motif analyses supported cCYP identification and phylogenetic classification. Moreover, based on these bioinformatics analyses, CYPs were suggested to be undergoing fast evolving gene systems, and each species has evolved multiple gene families with a diverse range of members. 

The CYP families are involved in xenobiotic metabolism, and are primarily expressed in liver tissue. Therefore, basal expression cartography of cCYPs in the liver of chickens at different developmental stages was constructed for the first time in our study. Our transcriptomic analysis revealed that cCYP2AC1, cCYP3A5, cCYP2C23a, cCYP4B7, and cCYP27A1 were highly expressed in chicken livers. Watanabe et al. reported that cCYP2C45 showed the highest mRNA expression in the livers of 8-week-old chickens [8]. In our results, cCYP2C45 was the second most abundant cCYP in 30-week-old chicken livers. Meanwhile, 11 cCYPs were differentially expressed between the pre-laying and laying hen livers. These data strongly indicated that mRNA abundance is thought to heavily depend upon chicken age. Moreover, during the hen laying cycle, most of the components of egg yolk, including triacylglycerols, free fatty acids and vitellogenin, are synthesized in the liver, and then transferred to the developing oocyte. Vigorous metabolic progress occurs in the liver of laying hens compared with the pre-laying hens. Thus, the significantly regulated cCYPs were suggested to be participating in a hepatic lipid metabolism process. 

Interestingly, a pair of duplicated genes, cCYP3A4 and cCYP3A5, showed substantially divergent expression profiles in chicken livers (Figure 4b). Watanabe et al. reported that cCYP3A5 (also known as CYP3A37) and cCYP3A4 (also known as CYP3A80) showed relatively low expression in 8-week-old chicken livers [8]. In our data, cCYP3A5 mRNA abundance was also relatively low in 10-, 20-, and 30-week-old chicken livers. However, cCYP3A4 mRNA abundance was greatest in 20- and 30-week-old chicken livers. Moreover, cCYP3A5 mRNA abundance in 30-week-old chickens was over five-fold higher than the second most abundant gene, cCYP2C45. The divergent expression profiles of cCYP3A4 and cCYP3A5 indicate that they are involved in different functional pathways in avian xenobiotic metabolism. Interestingly, the CYP3A4 and CYP3A5 genes were suggested to share a common regulatory pathway for constitutive expression in humans [50]. Our results also showed that chicken and human CYP3 subfamily genes were clustered into one branch in our phylogenetic tree (Figure 1). Further study is needed to elucidate the functional divergence of these homologous genes. 

CYP expression level was thought to be altered by various factors, such as age, sex, drugs, and environmental contaminants [8]. In this study, we further investigated mRNA induction by estrogen administration. Eleven cCYP members showed significant induction by estrogen, which indicates that cCYPs are involved in the estrogen signaling pathway in chicken livers. The CYP1 subfamily has been generally considered to be responsible for the hydroxylation and oxidation of 17β-estradiol in mammals [50,51]. However, no cCYP1 members responded to estrogen in chicken liver, which indicates that CYP1 showed species-specific regulation. The most significant difference was observed for cCYP2AC1, which was significantly downregulated over 15-fold by estrogen. Moreover, cCYP2AC1 was identified as the most abundant cCYP in 10-week-old chicken livers, and it was significantly downregulated five-fold in 30- compared with 20-week-old chicken livers. These data strongly indicate cCYP2AC1 suppression during the sexual maturity of chickens. Previously, avian CYP2AC1 was shown to be positively selected [17] and showed highly conserved synteny [8]. However, functions or physiological roles of cCYP2AC1 were not reported until now. Our study may provide an important foundation for future functional analysis of cCYP2AC1. 

## Figures and Tables

**Figure 1 genes-10-00617-f001:**
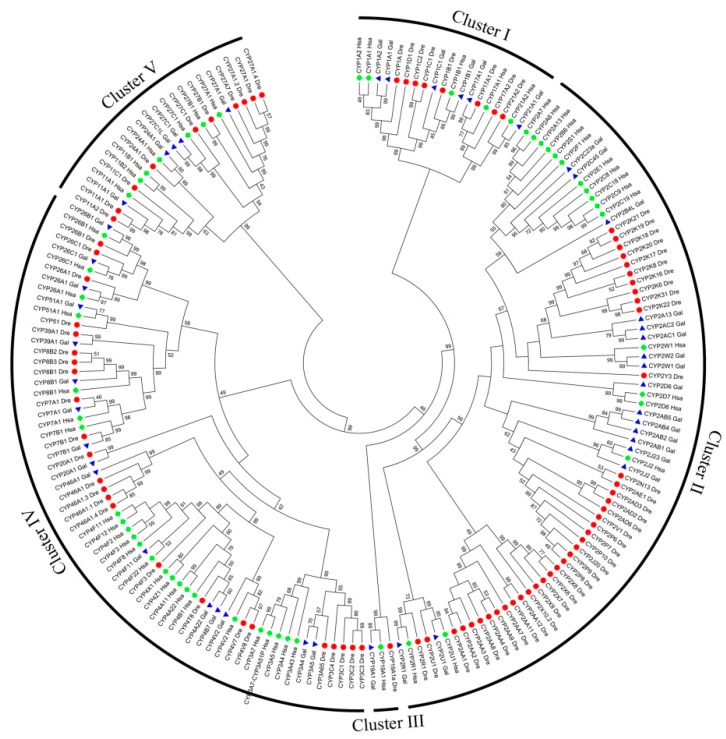
Phylogenetic tree of Cytochrome P450 (CYP) from human, chicken, and zebrafish genomes. The phylogenetic tree shows 57 CYPs from *Homo sapiens* (Has, highlighted in green rhombus), 45 CYPs from *Gallus gallus* (Gal, highlighted in blue triangles), and 83 CYPs from *Danio rerio* (Dre, highlighted in red circles). The phylogenetic tree was built in MEGA5 using the neighbor-joining method with 1000 bootstrap replicates.

**Figure 2 genes-10-00617-f002:**
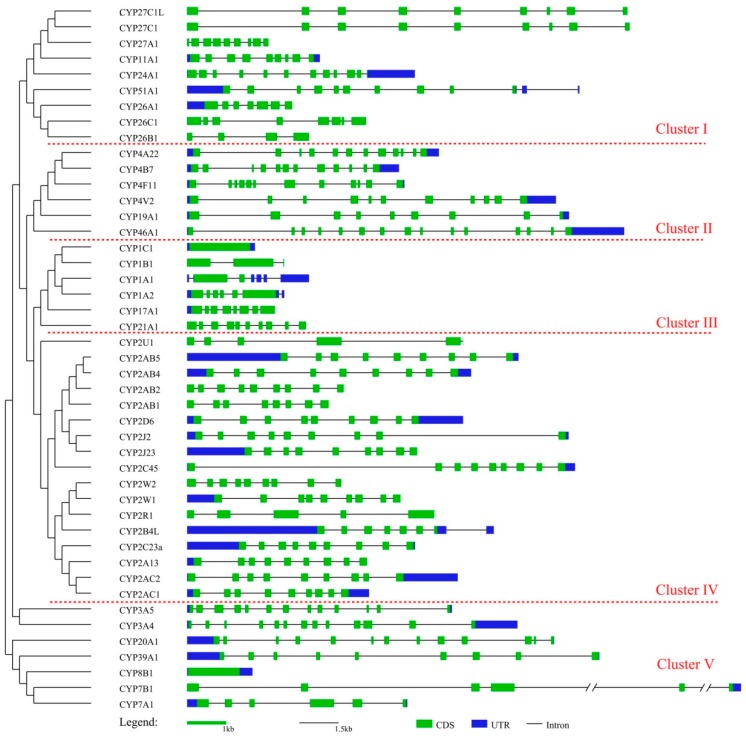
Chicken CYP (cCYP) phylogenetic relationships and gene structures. The phylogenetic tree was built in MEGA5 using the neighbor-joining method with 1000 bootstrap replicates. Gene structure analysis was carried out with the online tool GSDS 2.0. The length of the 5th intron in cCYP7B1 is 109,703 bp.

**Figure 3 genes-10-00617-f003:**
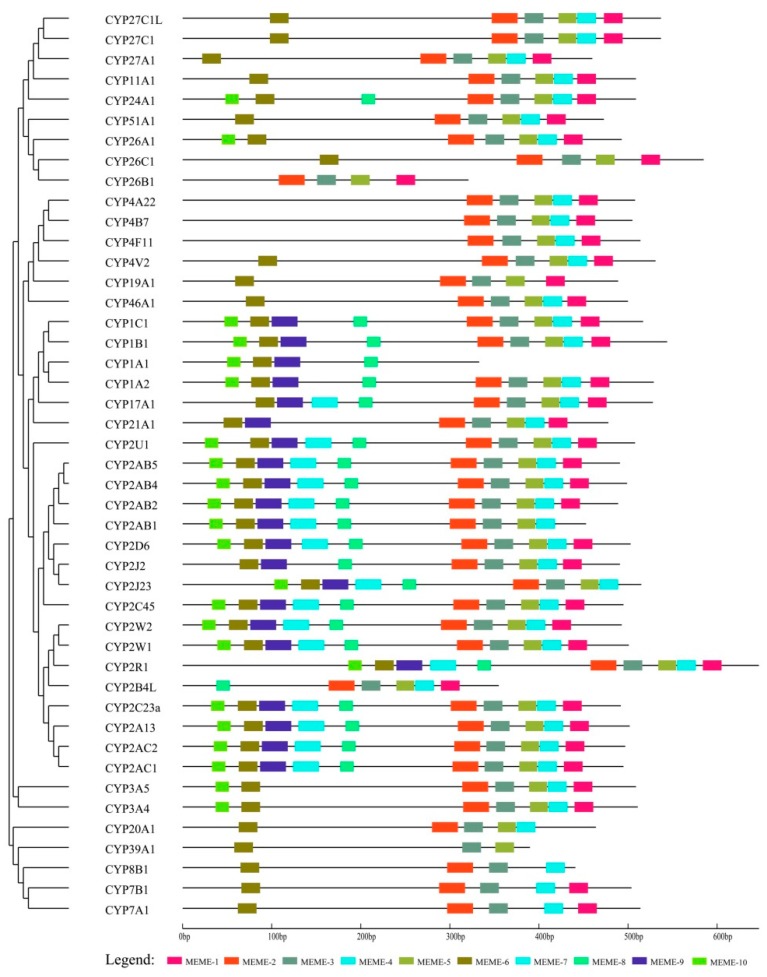
Conserved motifs of cCYPs. All motifs were identified by MEME with the complete amino acid sequences.

**Figure 4 genes-10-00617-f004:**
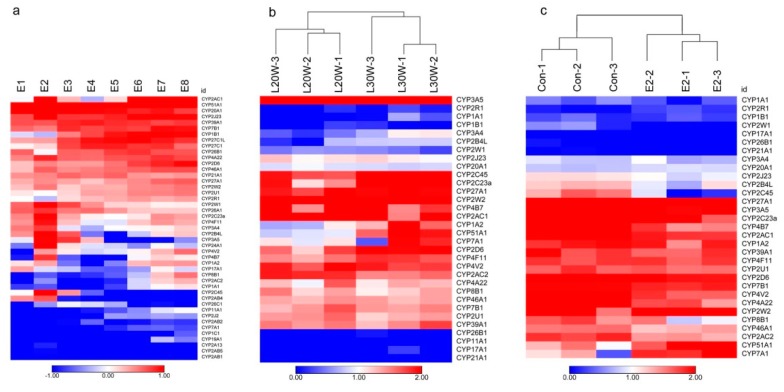
Heat maps showing the mRNA expression profiles of cCYPs. (**a**) Expression profiles of cCYPs in developmental stages of chicken embryos. (**b**) Expression profiles of cCYPs in the liver of pre-laying and laying hens. (**c**) Expression profiles of cCYPs in the liver of 17β-estradiol treated and non-treated chickens. The genes expression levels were obtained based on transcriptional data. The relative expression was log transformed and visualized as heat maps.

**Figure 5 genes-10-00617-f005:**
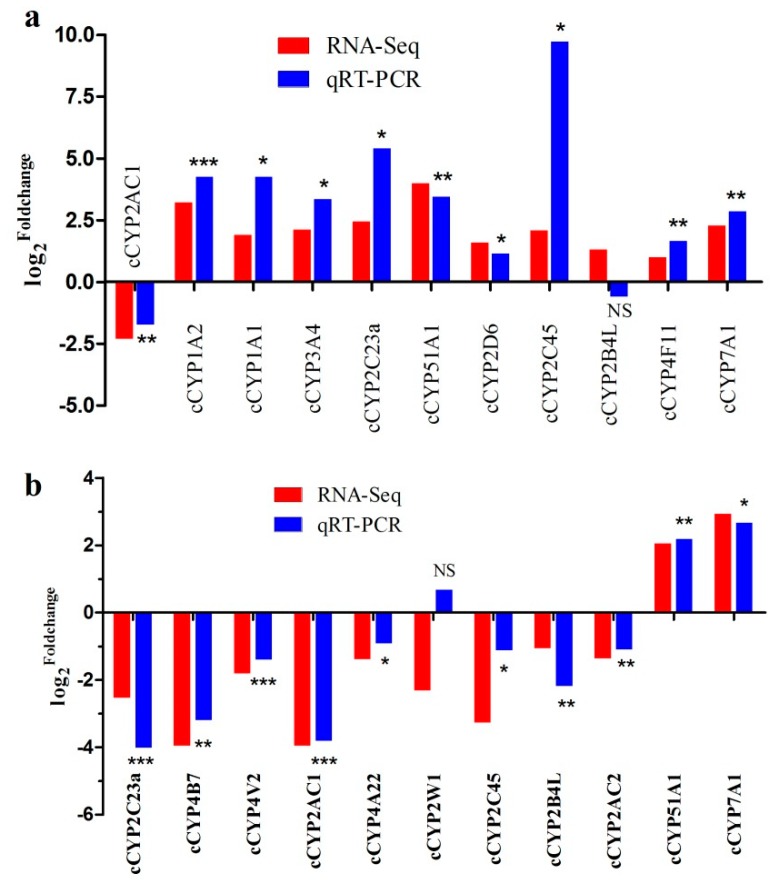
Quantitative real-time PCR (qRT-PCR) validation of differentially expressed genes. (**a**) log_2_^fold change^ of genes which were differentially expressed between 20 weeks old and 30 weeks old chicken liver. (**b**) log_2_^fold change^ of genes which were identified as estrogen responsive genes. The relative expression values of genes were calculated using the 2^−^^ΔΔCT^ method. Red indicates the log_2_^fold change^ of RNA-sequencing (RNA-seq) and green indicates the log_2_^fold change^ of qRT-PCR. Asterisks denote statistically significant differences of qRT-PCR (*t*-test; * *p* < 0.05; ** *p* < 0.01; *** *p* < 0.001; NS *p* > 0.05).

**Figure 6 genes-10-00617-f006:**
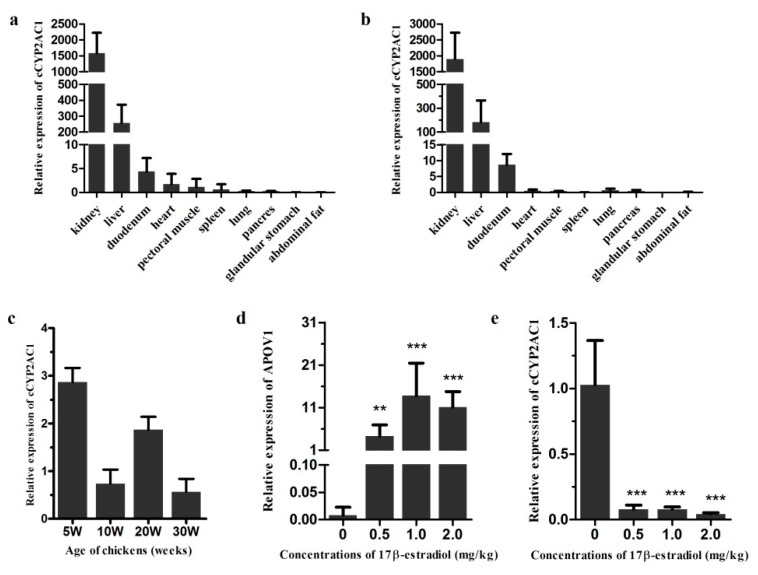
Expression characteristics and regulation of cCYP2AC1. (**a**) Tissue distribution of cCYP2AC1 in 10 week old chicken tissues. (**b**) Tissue distribution of cCYP2AC1 in 30 week old chicken tissues. (**c**) Expression patterns of cCYP2AC1 in livers of chickens at different developmental stages. (**d**) Effect of 17β-estradiol on APOV1 mRNA expressions in the liver of chickens. (**e**) Effect of 17β-estradiol on cCYP2AC1 mRNA expressions in the liver of chickens. Each data point represents the mean ± SD (*n* = 3–6; *t*-test; ** *p* < 0.01; *** *p* < 0.001; NS *p* > 0.05).

**Table 1 genes-10-00617-t001:** Differentially expressed genes between pre-laying and laying hens. Fragments per kilobase of exon per million mapped reads (FPKM).

Gene Name	Mean FPKM_20W	Mean FPKM_30W	Log2fold Change	FDR	UP/DOWN
CYP2AC1	454.764	92.029	−2.305	3.32 × 10^−3^	DOWN
CYP1A2	7.137	66.436	3.219	8.01 × 10^−7^	UP
CYP1A1	0.687	2.576	1.907	2.45 × 10^−4^	UP
CYP3A4	2.250	9.706	2.109	4.41 × 10^−8^	UP
CYP2C23a	41.463	226.648	2.451	1.48 × 10^−5^	UP
CYP51A1	7.207	114.250	3.987	5.26 × 10^−13^	UP
CYP2D6	37.170	112.042	1.592	1.31 × 10^−4^	UP
CYP2C45	59.638	253.827	2.090	3.68 × 10^−6^	UP
CYP2B4L	2.537	6.291	1.310	1.87 × 10^−2^	UP
CYP4F11	28.597	57.350	1.004	5.91 × 10^−4^	UP
CYP7A1	11.291	55.082	2.286	2.11 × 10^−2^	UP

**Table 2 genes-10-00617-t002:** Estrogen-responsive cCYPs in chicken liver.

Gene Name	Mean FPKM_Con	Mean FPKM_E2.0	Log2fold Change	FDR	UP/DOWN
CYP2C23a	502.082	86.824	−2.53176	7.96 × 10^−13^	DOWN
CYP4B7	618.130	39.801	−3.95704	1.45 × 10^−27^	DOWN
CYP4V2	172.665	49.238	−1.81013	3.77 × 10^−11^	DOWN
CYP2AC1	1027.453	66.303	−3.95385	1.47 × 10^−50^	DOWN
CYP4A22	110.614	42.371	−1.38437	4.54 × 10^−6^	DOWN
CYP2W1	2.976	0.594	−2.32607	1.80 × 10^−6^	DOWN
CYP2C45	33.088	3.414	−3.27673	1.67 × 10^−6^	DOWN
CYP2B4L	18.566	8.897	−1.06128	2.87 × 10^−2^	DOWN
CYP2AC2	62.489	24.216	−1.36764	5.24 × 10^−9^	DOWN
CYP51A1	21.543	89.010	2.046759	2.18 × 10^−5^	UP
CYP7A1	12.513	95.430	2.931051	1.72 × 10^−5^	UP

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
