# Peer review of "Global Investigation of Cytochrome P450 Genes in the Chicken Genome"

_genes, 2019, doi:10.3390/genes10080617_

Round 1

Reviewer 1 Report

Interesting paper. Just a few remarks -

line 55: CYP1A1 is generally considered a major CYP - this is not true statement, for example, for humans it is the CYP3A4 and so on - should be written as ....a major CYP in birds (or chicken), or.....considered as one of most important and conserved CYPs....

References: Please correct the Ref 31 (why capitals?)

Author Response

Point 1:

We have changed the "CYP1A1, which is generally considered a major CYP" to "which is considered as one of most important CYPs"  in the revised manuscript.

Point 2:

We have corrected the Ref 31.

Reviewer 2 Report

In this study, the chicken genome was analyzed to identify 45 chicken CYPs and their classification and evolutionary relationships were investigated by phylogenetic, conserved protein motif, and gene structure analyses. The differential expression of cCYPs in developing embryos revealed involvement of cCYPs in chicken development. Also 11 cCYPs, including cCYP2AC1, cCYP2C23a, and cCYP2C23b, were identified as estrogen-responsive genes. Overall this study provides evolutionary insight and helps elucidate their roles in physiological and toxicological processes in chicken. The work is done well with systematic manner, and manuscript is written well.

Minor Comments

In p11, this reviewer could not understand why [Figure 1] is there.

In p 8, P450 expression profiles of 20 weeks old of pre-laying hens and 30 weeks old laying hens were compared. The authors need to explain why they chose them and to discuss more about the results

Author Response

Point 1:

We have deleted these redundant sentences in the revised manuscript.

point 2:

Liver is the main metabolic organ in chicken. Most of the component of egg yolk including triacylglycerols, free fatty acids and vitellogenin are synthesized in the liver, and then transferred to the developing oocyte. Therefore, vigorous metabolic progress occurs in liver of laying hens compared with the pre-laying hens. 

Therefore, we chose 20 weeks old of pre-laying hens and 30 weeks old laying hens to investigate the potential role of cCYPs in hepatic metabolism.

According to your comments, we have added the information and discussion in lines 235-238 and lines 359-363.

Reviewer 3 Report

The authors identified in a very broad approach 45 cytochromes P450 in chicken. They were clustered and put in a phylogenetic tree together fish CYPs of human and zebrafish. RNA data from different embryonic stages of chicken revealed a differential regulation of some cCYPs in here. Futhermore, 11 cCYPs have shown differential regulation by 17β-estradiol administration.

The figures study seems to be conclusive, the figures are very good and the literature contains the relevant and newest publications.

Nonetheless, there were some points that limit my enthusiasm:

1st: The text hast to be edited by a native speaker or a language editing service.

2nd: The abstract should be rewritten. Background (Cytochrome P450 (CYP) superfamily enzymes are broadly involved in a variety of physiological and toxicological processes.), objectives (However, genome-wide analysis of this superfamily has never been investigated in the chicken genome. In this study, genome-wide analyses identified 45 chicken CYPs (cCYPs) from the chicken genome, and their classification and evolutionary relationships were investigated by phylogenetic, conserved protein motif, and gene structure analyses.) and general conclusion/outlook (These data expands our view of the phylogeny and evolution of cCYPs, and provides evolutionary insight, and can help elucidate their roles in physiological and toxicological processes in chicken.) are fine. But the summary of the results and a thematic conclusion need to be rewritten.

3rd: Results and discussion should be congruent. Either 198 or 184 CYPs were clustered. Please urgently check!

4th: Line 276/277 need a correction (Then we focused on the cCYP2AC1, which was one of the most changed genes in our transcriptomic data and not studied previously.) This was studies in literature [8] and [34].

Author Response

Point 1:

The revised manuscript had used english editing service from Edaz.

.

Point 2:

We have rewritten the abstract as follows:

Abstract: Cytochrome P450 (CYP) superfamily enzymes are broadly involved in a variety of physiological and toxicological processes. However, genome-wide analysis of this superfamily has never been investigated in the chicken genome. In this study, genome-wide analyses identified 45 chicken CYPs (cCYPs) from the chicken genome, and their classification and evolutionary relationships were investigated by phylogenetic, conserved protein motif, and gene structure analyses. The comprehensive evolutionary data revealed several remarkable characteristics of cCYPs, including the highly divergent and rapid evolution of the cCYPs, and the loss of cCYP2AF in the chicken genome. Furthermore, cCYP expression profile was investigated by RNA-seq. The differential expression of cCYPs in developing embryos revealed involvement of cCYPs in embryonic development. The significantly regulated cCYPs suggested its potential role in hepatic metabolism. Additionally, 11 cCYPs, including cCYP2AC1, cCYP2C23a, and cCYP2C23b, were identified as estrogen-responsive genes, which indicates that these cCYPs were involved in the estrogen-signaling pathway. Meanwhile, expression profile analysis highlight the divergent role of different cCYPs. These data expands our view of the phylogeny and evolution of cCYPs, and provides evolutionary insight, and can help elucidate their roles in physiological and toxicological processes in chicken.

Point 3:

A total of 184 CYPs were clustered in the study. We have corrected the mistake both in results and discussion (line 160).

Point 4:

We have corrected the sentence as follows: Then we focused on the cCYP2AC1, which was one of the most changed genes in our transcriptomic data.